# A Comparative and Critical Analysis for In Vitro Cytotoxic Evaluation of Magneto-Crystalline Zinc Ferrite Nanoparticles Using MTT, Crystal Violet, LDH, and Apoptosis Assay

**DOI:** 10.3390/ijms241612860

**Published:** 2023-08-16

**Authors:** Juan Luis de la Fuente-Jiménez, César Iván Rodríguez-Rivas, Irma Beatriz Mitre-Aguilar, Andrea Torres-Copado, Eric Alejandro García-López, José Herrera-Celis, María Goretti Arvizu-Espinosa, Marco Antonio Garza-Navarro, Luis Gerardo Arriaga, Janet Ledesma García, Domingo Ixcóatl García-Gutiérrez, Alejandro Zentella Dehesa, Ashutosh Sharma, Goldie Oza

**Affiliations:** 1Centre of Bioengineering, School of Engineering and Sciences, Tecnologico de Monterrey, Campus Queretaro, Av. Epigmenio González, No. 500, Fracc. San Pablo, Querétaro 76130, Mexico; juan.luis.delafuente@tec.mx (J.L.d.l.F.-J.); a01700238@tec.mx (A.T.-C.); goretti@tec.mx (M.G.A.-E.); 2División de Investigación y Posgrado, Facultad de Ingeniería, Universidad Autónoma de Querétaro, Cerro de Las Campanas S/N, Querétaro 76010, Mexico; cesarmediuaq@gmail.com (C.I.R.-R.); janet.ledesma@uaq.mx (J.L.G.); 3Unidad de Bioquímica, Instituto Nacional de Ciencias Médicas y Nutrición Salvador Zubirán, Vasco de Quiroga 15, Belisario Domínguez secc 16, Tlalpan, Mexico City 14080, Mexico; irma.mitrea@incmnsz.mx (I.B.M.-A.); garcialp007@gmail.com (E.A.G.-L.); 4Centro de Investigación y Desarrollo Tecnológico en Electroquímica, Parque Tecnológico, Sanfandila, Pedro Escobedo, Querétaro 76703, Mexico; jlherrera@cideteq.mx (J.H.-C.); larriaga@cideteq.mx (L.G.A.); goza@cideteq.mx (G.O.); 5Facultad de Ingeniería Mecánica y Eléctrica, FIME, Universidad Autónoma de Nuevo León, UANL, Av. Universidad S/N, Cd. Universitaria, San Nicolás de los Garza 66450, Mexico; marco.garzanr@uanl.edu.mx (M.A.G.-N.); domingo.garciagt@uanl.edu.mx (D.I.G.-G.); 6Departamento de Medicina Genómica y Toxicología Ambiental, Programa Institucional de Cáncer de Mama, Instituto de Investigaciones Biomédicas IIB & Red de Apoyo a la Investigación, Universidad Nacional Autónoma de México, Ciudad Universitaria, Ave. Universidad 3000, Col. Copilco Universidad, Del. Coyoacán Ciudad de México, Mexico City 04510, Mexico

**Keywords:** zinc ferrite, co-precipitation, magneto-crystalline, triple-negative breast cancer, cytotoxicity, MTT, LDH, flow cytometry, apoptosis

## Abstract

Zinc ferrite nanoparticles (ZFO NPs) are a promising magneto-crystalline platform for nanomedicine-based cancer theranostics. ZFO NPs synthesized using co-precipitation method are characterized using different techniques. UV-visible spectroscopy exhibits absorption peaks specific for ZFO. Raman spectroscopy identifies Raman active, infrared active, and silent vibrational modes while Fourier transforms infrared spectroscopic (FTIR) spectra display IR active modes that confirm the presence of ZFO. X-ray diffraction pattern (XRD) exhibits the crystalline planes of single-phase ZFO with a face-centered cubic structure that coincides with the selected area electron diffraction pattern (SAED). The average particle size according to high-resolution transmission electron microscopy (HR-TEM) is 5.6 nm. X-ray photoelectron spectroscopy (XPS) signals confirm the chemical states of Fe, Zn, and O. A superconducting quantum interference device (SQUID) displays the magnetic response of ZFO NPs, showing a magnetic moment of 45.5 emu/gm at 70 kOe. These ZFO NPs were then employed for comparative cytotoxicity evaluation using MTT, crystal violet, and LDH assays on breast adenocarcinoma epithelial cell (MCF-7), triple-negative breast cancer lines (MDA-MB 231), and human embryonic kidney cell lines (HEK-293). Flow cytometric analysis of all the three cell lines were performed in various concentrations of ZFO NPs for automated cell counting and sorting based on live cells, cells entering in early or late apoptotic phase, as well as in the necrotic phase. This analysis confirmed that ZFO NPs are more cytotoxic towards triple-negative breast cancer cells (MDA-MB-231) as compared to breast adenocarcinoma cells (MCF-7) and normal cell lines (HEK-293), thus corroborating that ZFO can be exploited for cancer therapeutics.

## 1. Introduction

Spinel zinc ferrites (ZnFe_2_O_4_ or ZFOs) are magneto-crystalline nanoparticles (MCNs) with several properties and applications in biomedicine and nanotechnology. They have a great demand in drug delivery, tissue engineering, and hyperthermia, and act as a contrast agent in magnetic resonance imaging due to their stability in colloidal suspensions, non-immunogenicity, non-toxicity, and biocompatibility [1,2]. Hence, ZFOs have vastly contributed to biomedical functions mostly due to their biocompatible nature [3]. Moreover, spinel ferrites have a structural formula (A) [B]_2_O_4_ (also known as MFe_2_O_4_) in which (A) is a divalent metal cation from 3D transition metals such as zinc, copper, cobalt, and nickel, among others, and possesses availability for cations at the tetrahedral site, while [B] occupies the octahedral sites. These spinel ferrites are in a unit cell of 32 oxygen atoms of a cubic-closed packed arrangement. Zinc ferrite NPs have a normal spinel ferrite structure in which all the Zn^2+^ ions are in the A site and Fe^3+^ ions enter in the B sites [4,5]. The most common structure of spinel ferrites is the inverse structure, in which half of the Fe^3+^ ions stay in the octahedral site and the other half replaces the (A) cations in the tetrahedral site [4]. This imparts a strong paramagnetic behavior, since Zn^2+^ ions do not have a magnetic moment in the tetrahedral site. However, nanocrystalline ZFO NPs have an inverse spinel structure in which the Zn^2+^ and Fe^3+^ ions are partially dispersed in both the sites having antiferromagnetic behavior [4,5,6]. In addition, MNPs produced from iron oxide are composed of a core either from maghemite (γ-Fe_2_O_3_) or magnetite (Fe_3_O_4_) and coated with biocompatible layers. Magnetite is the most common crystalline phase of iron oxide for biomedical purposes [2]. Furthermore, MNP synthesis can be performed with several procedures such as thermal decomposition, sol-gel, hydrothermal synthesis, microemulsion, and co-precipitation [2]. From all the above-mentioned synthesis procedures, co-precipitation has several advantages, such as water dispersibility, facile route of functionalization, is the most employed synthesis in biomedical applications due to its hydrophilicity and is an accessible and low-cost large-scale manufacturing practice. NPs obtained from this procedure have high purity suitable for magnetic characterization [2,4]. During co-precipitation synthesis, aqueous solutions of salts from Fe^2+^ and Fe^3+^ are coprecipitated after the addition of a base. The control of ZFO composition, shape, and size changes according to the Fe^2+^ and Fe^3+^ ratio used, ionic strength of the medium, and pH [2].

In this present study, ZFO NPs were synthesized using a co-precipitation method. The characterization of the structural composition, size, and morphology by UV-visible spectroscopy, X-ray diffraction (XRD), Fourier transform infrared spectroscopy (FTIR), high-resolution transmission electron microscopy (HR-TEM), superconducting quantum interference device (SQUID), X-ray photoelectron spectroscopy (XPS), and RAMAN spectroscopy was performed. Additionally, the cytotoxicity of the ZFO NPs was assessed against triple negative human breast cancer cell line MDA-MB-231 (defined as tumors that lack expression of estrogen, progesterone receptor, and human epidermal growth factor [7]), human breast cancer cell line (MCF-7), and human embryonic kidney 293 (HEK-293) cell lines with different cytotoxicity assays such as 3-(4,5-dimethylthiazol-2-yl)-2,5 diphenyltetrazolium bromide (MTT), crystal violet, and lactate dehydrogenase (LDH). In addition, a count of cells entering early and late apoptosis was measured with flow cytometry. The main goal of this article was to compare the assays for studying cytotoxicity or viability to avoid any interference, leading to under- or over-estimation of results for the proper comprehension of the ZFO effect.

## 2. Results and Discussion

The mechanism of ZFO NP synthesis based on co-precipitation reaction is as follows [5]: In the first step, as in Reaction 1 below, there is an interaction between Fe^3+^ and OH^−^, which moves in the forward direction to form an increasing concentration of FeOOH that possesses low solubility. Reaction 2 is a slow-and-rate determining step in which the intermediate FeOOH reacts with zinc ions at an alkaline pH to form zinc ferrite, spontaneously.
Fe^3+^ + 3OH^−^ → FeOOH + H_2_O(1)
FeOOH + Zn^2+^ + 2OH^−^ → ZnFe_2_O_4_ + 2H_2_O(2)

### 2.1. Optical and Molecular Vibrational Characterization of ZFO NPs

The UV-visible absorption spectroscopy is a characterization technique employed to determine the electronic structure and optical properties of the NPs. The obtained absorption peaks are normally related to the diameter, shape, and size of the NPs [8]. Figure 1A shows the UV-visible spectra of ZFO solid nanopowder in the range of 200–800 nm at room temperature. These ZFO NPs are photoresponsive in both UV and visible light ranges, especially in the wavelength ranging from 300 to 600 nm [9]. In Figure 1, the predominant absorption features of ZFO NPs are noticed at 430, 500, and 550 nm. The evidence suggests optical transitions around 430 and 500 nm that are assigned to (^6^A_1g_ → ^4^T_2g_(D)) and (^6^A_1g_ → ^4^A_1g_(G)) transitions, respectively. The 550 nm band corresponds to O^2−^ + Fe^3+^ → O^−^ + Fe^2+^ indirect transition [10]. Likewise, the obtained dark brown color of ZFO proved the visible light absorption capacity of these NPs [8,10].

Raman spectra in Figure 1B corroborates the structure and arrangement of the cubic phase of ZFO NPs. Appendix A shows the microscopic image of ZFO NPs. The space group of the cubic structure of ZFO NPs belongs to the O^7^_h_ (*Fd-3m*) space group with eight molecules per unit cells ending in 56 atoms; however, the lowest Bravais cell includes two formula units for a total of 14 atoms. According to the factor group theory it is possible to derive the next 42 modes of ZFO NPs spinel with 3 acoustic and 39 optical modes at the Γ-point of the Brillouin zone [6,11]:A_1g_ (R) + E_g_ (R) + F_1g_ + 3F_2g_ (R) + 2A_2u_ + 2E_u_ + 4F_1u_ (IR) + 2F_2u_
where the (IR) and (R) identify infrared active and Raman active vibrational species, respectively, while the rest of the modes are silent modes [6,12]. Thus, the five subsequent vibrational modes of ZFO NPs at room temperature anticipated for spinel with a cubic structure are A_1g_, E_g_, and 3F_2g_ Raman active while the rest are silent modes, except F_1u_ which is infrared-active [3,6]. The three Raman-active modes of 3F_2g_ are identified as F_2g_ (1), F_2g_ (2), and F_2g_ (3), F_2g_ (1) being the mode with the lowest frequency and F_2g_ (3) the mode with the highest frequency of the vibrational species [6,12]. A visual review of the spectra shows several Raman bands in Figure 2 which can be categorized according to the mode region. The A_1g_ mode region is between 600–800 cm^−1^, in which a band at 670 cm^−1^ can be observed; in the F_2g_ (3) mode region between 410–550 cm^−1^, two bands at 459 cm^−1^ and 501 cm^−1^ can be identified; and in the F_2g_ (2) mode region between 260–380 cm^−1^, bands at 316 cm^−1^ and 373 cm^−1^ were detected. At lower frequencies, the E_g_ and F_2g_ (1) modes are occasionally detected (Figure 2) [6,12]. The remaining Raman bands at higher energies, between 1000 and 1800 cm^−1^, can reveal second-order spectral characteristics [6]. In Figure 2, several second-order features can be noticed, such as 1061 cm^−1^, 1137 cm^−1^, 1173 cm^−1^, 1327 cm^−1^, and 1538 cm^−1^. Nevertheless, the 1337 cm^−1^ could also be linked to hematite structures [13].

Furthermore, it is known that in cubic spinel such as ferrites, lower frequency modes are attributed to metal–oxygen vibrations in the octahedral BO_6_ sites, while frequencies above 600 cm^−1^ correspond to metal–oxygen vibration in tetrahedral AO_4_ sites. Therefore, it is implied that zinc cations are favorably located in tetrahedral A sites, as there is a shift in the A_1g_ mode as zinc is integrated in the structure [3,12]. Hence, the mode at 670 cm^−1^ is from the Zn–O vibration which is a predominant peak approximately in 650 cm^−1^ at the tetrahedral site and the rest is from the Fe–O octahedral vibration site (Figure 2) [3]. Finally, the Raman frequency varies on the bond length of Fe (Zn)–O which fluctuates with the particle size and lattice parameter variation [3]. In addition, zinc ferrite NPs have a partially inverted spinel structure which suggests a modification of the allocation of Fe^3+^ and Zn^2+^ ions between the octa- and tetrahedral sites. Thus, at the same time, two categories of cations can vibrate guiding two distinctive Raman modes with a very near frequency, and consequently, the appearance of broad peaks in the spectra are generally perceived [11,12]. This could be the reason for the proximity of peaks 316 cm^−1^ and 373 cm^−1^, 459 cm^−1^ and 501 cm^−1^, and 1137 cm^−1^ and 1173 cm^−1^ (Figure 2).

FTIR spectra aids in confirming the spinel structure of ZFO NPs, which is shown in Figure 1C. The normal spinel ferrites are known to display four IR active modes (*v*_1_, *v*_2_, *v*_3_, *v*_4_,) in the vibration spectra. The high frequency bands (*v*_1_ and *v*_2_) are very sensitive to interaction between cations and oxygen in the tetrahedral (A-site) with stretching vibration of the metal–oxygen bonds and octahedral (B-site) with the bending vibration of the metal–oxygen bond positions. Normally, A-site appears between 540–600 cm^−1^ and B-site between 400–450 cm^−1^ [8,13]. Additionally, high frequency bands around 667 cm^−1^ are attributed to tetrahedral groups and the position of the characteristic bands robustly depends on the structural changes of the NPs, such as size of the powder grain, atomic mass, and cation radii, making them sensitive to any alterations when there is interaction between cations and oxygen in the tetra- and octahedral positions [8,13]. According to the possible interaction change between the tetra- and octahedral position, in this study the absorption band at 419 cm^−1^ is related to a bending vibration of Fe-O in the octahedral site, while the band at 621 cm^−1^ is related to the Zn-O stretching vibration in the tetrahedral site, both created by the interaction of metal ions with oxygen [13,14,15,16]. This 621 cm^−1^ Zn-O vibration band in the tetrahedral site is consistent with the results of the RAMAN spectroscopy (Figure 2). Furthermore, peaks at 831 cm^−1^ are linked to vibration frequencies of the organic functional groups in the NPs corresponding to C-O stretching and bending vibrations, which in this study correspond to the 853 cm^−1^, and the peaks around 1105 cm^−1^ are related to the Fe-M ferrite system linked to the obtained peak at 1060 cm^−1^ [17,18,19]. In addition, two types of O-H interactions with zinc ferrite NPs are suggested. Type 1 is linked to the bending vibrations of the absorbed water at 1520 cm^−1^ and Type 2 bending vibration caused by a synchronized hydrogen absorption at both Zn and O sites, yielding a peak at 1327 cm^−1^ [20]. These peaks are exhibited in Figure 3, in which approximate absorption peaks at 1344 cm^−1^ and 1516 cm^−1^ can be seen, implying an interaction with water. These peaks could be related to absorbed water molecules or humidity, since a strong and sharp absorption peak at approximately 1650 cm^−1^ is assigned to stretching and bending vibrations of O-H functional groups on the surface of zinc ferrite NPs, which in this case resembles to the strong and sharp peak obtained at 1516 cm^−1^ [13,21]. Moreover, the presence of these absorption peaks and other strong and broad absorption peaks between 3200–3600 cm^−1^ are attributed to hydrogen bonded stretch vibration of O-H functional groups [14]. These absorption peaks exhibit H-O-H and O-H vibrations, confirming the presence of water molecules adsorbed on the surface of the NPs caused by the specific area of the ZFO nanopowders [14]. Hence, the band of 3178 cm^−1^ may be due to O-H stretching vibration groups of adsorbed water on the surface of the ZFO NPs (Figure 3) [4,18,22].

### 2.2. Structural Characterization of ZFO NPs

Figure 1D shows the result for the crystalline phase and structure of the ZFO NPs. The observed peaks in the XRD diffraction patterns and their intensity and position are very similar to the standard powder diffraction for the cubic spinel structure of zinc ferrite [23]. The diffraction peaks at 2θ values of ZFO NPs shown in Figure 4 are 30°, 35.36°, 39.2°, 41.88°, 46.8°, 52.84°, 57.44°, 62.68°, 75.52°, 79.24°, and 80.12°, which are in strong agreement with the crystalline planes of (220), (311), (222), (400), (331), (422), (511), (440), (622), (444), and (551), respectively, from the cubic system of zinc ferrite JCPDS pattern no. 01-089-4926 and 00-022-1012, confirming the presence of a single ZFO NPs phase with a face-centered cubic structure with space group *Fd3m*. The most intense peak of ZFO nanopowder is (311) [2,21,23,24]. Moreover, sodium hydroxide of 1.5 M or above, as well as 80 °C used during the single-phase ZFO NPs synthesis method, produces stronger, narrower, and sharper diffraction peaks. This is also caused by the rise of crystallinity, crystallite size, and small size of the NPs, which can be seen in the peaks of Figure 4 [4,5,21]. On the other hand, some minor peaks related to the impurities of Fe_2_O_3_ phases (hematite) were detected in the sample; these impurities happen naturally or due to a possible incomplete reaction during the co-precipitation synthesis [23,25]. XRD peaks of iron oxide Fe_2_O_3_ were also detected at 24.84° (210), 25.45° (211), 49.52° (421), and 67.8° (442), according to JCPDS pattern no. 00-039-1346 (Figure 4). The reasons behind the appearance of these peaks are that ZFO NPs having Fe^3+^ as the main element and its manifestation signifies that there were more than 50 mole percent of the normal Fe_2_O_3_ content of spinel ferrite [22]. Moreover, one of the reasons for the hematite formation is due to the energy deficiency for attaching the Zn^2+^ ions with iron oxide to determine the crystal structure [22]. Finally, these impurities could be the result of using temperatures below the calcination range of 800 °C, since the co-precipitation synthesis method involves synthesis of NPs at low temperatures without the necessity of calcination at higher temperatures [21,26]. Furthermore, since the crystalline planes of ZFO NPs are properly indexed to a cubic spinel structure of the zinc ferrite, the crystallite size of the ZFO was estimated. An average crystallite size was calculated by employing the Scherrer’s equation over all the diffraction peaks:D = (0.9λ)/(βcosθ)
where λ is the wavelength of the X-ray radiation (λ = 0.154 nm), β is the full width at half-maxima of the peak in radians corrected for instrumental broadening, θ is the Bragg diffraction angle in radians, and D is the mean crystallite size [5,27]. The obtained average particle size was found to be approximately 5.6 nm for the NPs. This size is consistent with the HR-TEM results that confirm the size of ZFO NPs between 2.5 and 5 nm (Figure 4), as will be discussed in the following section. Additionally, MNPs typically consist of an iron oxide core between 5 to 20 nm, which is accomplished during the co-precipitation synthesis [1]. This size is correlated well since, as the NPs enter the blood circulation, they can be identified by the reticuloendothelial system (RES) and correspondingly eliminated from the vascular system. In the case of a particle size above 200 nm, the ability to enter tumor tissues decreases significantly and they are more vulnerable to suffer phagocytosis, thus leading to RES clearance. Hence, the size of the particle is crucial to improve the endocytic uptake by cells and their accumulation in tumor areas. For an effective cancer treatment, the usual size of the drug delivery system should be below 100 nm [1,28,29].

Figure 2 shows the TEM characterization results for the ZFO NPs sample to elucidate its morphology. In this figure, ZFO NPs were observed to form agglomerates (Figure 2a,b), showing dimensions of hundreds of nm to microns in size. These agglomerates were formed by “smaller” ZFO NPs with sizes between 2.5 nm and 5 nm, approximately, in close agreement with the calculated result by the Scherrer formula. In addition, the size distribution of the ZFO NPs is plotted in Figure 2a inset. The results showed a polydispersed size distribution with a ZFO NP mean size of 3.84 nm. These results agree with the HR-TEM and XRD average sizes, ranging from 2.5 and 5.6 nm. Furthermore, HR-TEM images (Figure 2c,d) of these NPs showed their crystalline nature. In these “small” ZFO NPs, a clear interplanar distance of ~0.24 nm could be observed, possibly related to the (222) planes in the ZFO NP cubic phase.

Moreover, Figure 3a shows the Selected Area Electron Diffraction (SAED) pattern of the ZFO sample with clear diffraction spots corresponding to the interplanar distances related to the crystal planes (220), (222) (400), (511), (440), and (444) in the ZFO NPs cubic phase (pattern no. 00-022-1012). These peaks coincide with the planes identified in the XRD characterization, confirming the interplanar distances observed. Additionally, Energy Dispersive X-Ray Spectroscopy (EDXS) spectra shown in Figure 3b and the table shown in the inset of 3b confirmed the composition of ZFO NPs to be Zn, Fe, and O. Nonetheless, in order to elucidate the chemical state of the identified elements, we proceed to record the high resolution Fe2p, Zn2p, and O1s XPS spectra.

Figure 4 displays the X-ray photoelectron spectroscopy showing a high resolution Fe2p, Zn2p and O1s spectra recorded for the powdered ZFO NPs and Appendix A shows complete survey scan of ZFO NPs (Appendix A). Accordingly, the Fe2p spectrum shows peaks at 710.8 and 724.5 eV related with core emissions from 2p_3/2_ and 2p_1/2_ states, respectively, for Fe(III) cations. In addition, there are satellite peaks observed at 718.6 and 734.3 eV which are attributed to shake-up-like emissions from Fe(III) cations [30]. Moreover, the Zn2p spectrum displays signals related to photoelectrons emitted from the 2p_3/2_ and 2p_1/2_ states that are located at 1021.2 and 1044.2 eV, respectively, which are congruent with those expected for Zn(II) cations [30]. Finally, the O1s spectrum can be deconvoluted into four peaks at 529.8, 531.3, 532.9 eV, and 535.0 eV associated with the (a) lattice oxygen in ZFO, (b) surface adsorbed organic molecules, (c) humidity, and (d) Auger electrons from Na, respectively. From these signals, we are assured that the chemical state of Fe, Zn, and O agrees with the expected chemical composition of as-synthesized spinel ferrite.

### 2.3. Magnetic Characterization of ZFO NPs

Figure 5 shows the m(H) curves recorded at 5 and 312 K for the ZFO NPs sample. As it can be noticed, this sample displays a magnetic response that fits with that expected from a soft ferromagnet, such as remanence (m_R_ = 5.9 emu/g), and coercivity (H_C_ = 0.33 kOe) at 5 K (see inset in Figure 5a). It is also observed that the magnetic moment reaches m_max_ = 45.5 emu/g at 70 kOe, although the m(H) does not show saturation even at this high applied field (Figure 5a). This lack of saturation can be attributed to the existence of a spin-glass-like surface layer in ZFO NPs not contributing to the magnetic response of the sample even at high magnetic fields, as it has been proposed for other spinel oxide NPs [31]. In addition, it is noticed that there is lack of hysteretic characteristics in the m(H) curve measured at 312 K (Figure 5b). This feature can be understood as the magnetic response of a ferromagnet in the superparamagnetic regime. Accordingly, above certain critical temperatures, known as blocking temperature (T_B_), the thermal energy provokes the relaxation time of spins to become smaller than the relaxation time of the measuring apparatus, hence, their magnetic response resembles that of a Langevin’s paramagnet. Figure 6 shows the m(T) curves recorded under zero field-cooled (ZFC) and field-cooled (FC) conditions. The ZFC curve displays a typical deblocking response of the spins as temperature increases, showing a maximum at T_M_ = 59.2 K. Above this temperature, the measured magnetic moment starts to decrease as the temperature moves towards 312 K. This diminishing magnetic moment can be attributed to the entry of the sample into the superparamagnetic regime; hence, T_M_ can be a rough estimation of its correspondent T_B_. Moreover, it is also observed that the FC curve diverges from the ZFC one at T_IRR_ = 154.7 K and depicts an increase in the magnetic moment as the temperature decreases. In fact, the magnetic moment reaches a magnitude higher than that recorded at T_B_. This result implies that additional blocked spins contribute to the magnetic response of sample during the FC process. These spins could be those in spin-glass-like surface layer in NPs that were identified from the measuring of the m(H) curves, but also a cooperative blockage of interacting nanoparticle from the observed clusters in the TEM images.

### 2.4. Comparative Cytotoxicity Evaluation Using Different Viability Assays (MTT, Crystal Violet and Lactate Dehydrogenase)

After a cautious characterization and estimation of the stability of nanoparticles, a biocompatibility and safety assessment is desirable for the NPs, beginning with in vitro assays for biomedical purposes. However, NPs can interfere with the cytotoxicity assays, leading to under- or overestimation of the NPs effect. Some examples of interferences are interaction of NPs with one of the components of the assay altering the functions of NPs or cellular metabolism, enhancing the optical activity of the reagents, or adsorbing the reagent on the surface of the NPs [32]. Therefore, there is a precise need for analyzing the cytotoxicity of nanoparticles with different assays and their combinations will give a more reliable result. To evaluate the cytotoxicity of the ZFO NPs, there were three viability assays that were tested, which are as follows:

#### 2.4.1. MTT Assay

The standard MTT assay was performed on the MDA-MB-231, MCF-7, and HEK-293 cell lines after 48 h incubation with the ZFO at several concentrations (4, 12, 20, 50, 100, 200, 300, 400, 500, and 600 μg/mL). The viability profiles of the treated cells are shown in Figure 7. Likewise, MTT evaluates the mitochondrial activity exhibited by the viable cells after the treatment. The major contribution of NPs towards cytotoxicity is the formation of reactive oxygen species (ROS) causing oxidative stress. ROS produces several radicals such as superoxide anions, hydrogen peroxide, as well as hydroxyl and alkoxy radicals which damage lipids, DNA, and proteins, among others [15]. In addition, the interaction of ZFO NPs with the cell membrane causes a rupture and deformation of the membrane, generating ROS and particle dissolution, following the release of free intracellular Zn^2+^ and subsequently leading to failure of the cellular redox system and thus causing ROS production inside the cell [15]. In Figure 7A, it can be observed that the most damaged cell line at lower concentrations of ZFO NPs is MCF-7, between 4 and 50 µg/mL while MDA-MB-231 has higher cell viability but is lower than untreated cells. Moreover, in some cases at higher concentration such as 100, 200, 400, and 500 µg/mL, MCF-7 exhibits more cell viability than the MDA-MB-231 cell line, but at 300 and 600 µg/mL MDA-MB-231 has higher cell viability. Hence, a variation of cell viability exposed to different ZFO NPs concentration among the two different breast cancer cell lines creates doubts about which cell line is more sensitive to the exposure of zinc ferrite NPs. Nevertheless, the HEK-293 cell line exhibits statistically similar or higher cell viability in comparison to the untreated cells in almost all the ZFO NP concentrations, suggesting that ZFO NPs have a good safety profile, showing more than 75% of viability in almost all the cases.

Nonetheless, based on the TEM, XRD, and HR-TEM results, the size of the NPs are very small, in the range of 2.5 to 5.6 nm unravels certain exclusive properties of NPs, as size-dependent surface reactivity rendering higher ambiguities in toxicological NP evaluation. There is a critical need to evaluate different assays and confirm the effect of the size of NPs, considering different assays may react differently to small-sized particles [33]. Hence, the reduced cell viability of HEK-293 at 4, 300, and 600 µg/mL could be inferred from the NP size, since the calculated size is an approximation around 5 nm and an agglomeration could have formed [34]. The higher cell viability at 12 and 500 µg/mL in the HEK-293 cell line could be either due to the absorption of the substrate, interference of Zn^2+^ with the reduction reaction of MTT, interaction with the substrate depleting free MTT and creating false negative results, or oxidation of MTT to formazan due to the high NP concentration [34,35]. Therefore, other cytotoxic assays need to be done for reducing this existing variance be-tween breast cancer cell lines and cell viability.

#### 2.4.2. Crystal Violet Assay

The crystal violet assay was employed with the same ZFO NPs concentrations as used in the MTT assay. These results confirm that the ZFO NPs damage more cancerous MDA-MB-231 and MCF-7 cell lines than the normal HEK-293 cell lines, as can be seen in the standard deviation presented in the graphs (Figure 7B). In this case, in all exposed concentration, the most damaged breast cancer cell line was MDA-MB-231 compared to MCF-7. This result creates a discrepancy with the obtained MTT results between which breast cancerous cell line is more sensitive to the several ZFO NP concentrations. However, HEK-293 cell line still exhibits a higher cell viability for nanoparticle treated cells as compared to the untreated ones, especially in some concentrations such as 4, 12, 20, 50, 100, 200, 300, and 400 µg/mL, which could be caused by the size, shape, aggregation, optical and magnetic properties, adsorption capacity, surface charge, and solubility, among others [31,35], as can be noted from the characterizations performed in this work. Nevertheless, an interesting result is that the HEK-293 cell line exhibited a reduced cell viability in both MTT as well as crystal violet assay at the same concentrations of ZFO NPs. Moreover, NPs may aggregate at the bottom of the culture plate and sediment affecting the outcome of cell viability due to the mass composition and concentration and spectrum absorbance, leading to a false interpretation [31,36,37]. In addition, MNPs are known to enhance the signal due to electrical interactions, or absorb UV-visible light, inducing a false positive viability result [31,34]. However, the results of this assay reinforce the suggestion that ZFO NPs showed a good safety profile in non-cancerous cell lines exhibiting more than 90% of viability in nearly all the cases. Therefore, another cytotoxic assay was performed to obtain a clear picture of the ZFO NP cytotoxicity and determine as to which breast cancer cell line was more sensitive and strengthen the fact that ZFO NP has a good safety profile against benign cells. 

#### 2.4.3. Lactate Dehydrogenase Assay

Thus, the Lactate dehydrogenase assay (LDH) realizes the cytotoxicity of NPs by measuring the extracellular content of released lactate dehydrogenase from damaged cells [33]. In this study, the ZFO NP exposure in all cell lines demonstrate LDH release in nearly the same ZFO NP concentration as observed in the MTT and crystal violet assay, except for 12 and 20 µg/mL concentrations. In this dose-response curve, the LDH concentration in the MDA-MB-231 cell line increases at lower concentrations such as 4, 50, 100, and 300 µg/mL as compared to MCF-7. This indicates a rise in dead cells, while at higher concentrations 200, 400, 500, and 600 µg/mL MDA-MB-231 showed a decrement in the LDH release and MCF-7 exhibited a higher LDH concentration resulting in more dead cells, thus switching positions with MDA-MB-231 for more dead cells (Figure 7C). This reduced quantity of LDH release could be due to the possibility of ZFO NP interaction with the LDH enzyme, thus causing its inactivation. One more possibility is that MNPs create a metal-catalyzed oxidation of LDH producing ROS and causing a site-specific damage to proteins [33]. Another possibility could be that the NPs could adsorb the LDH molecules on their surface, inactivate them, and then either release them and adsorb new LDH molecules (dynamic adsorption) or attach them strongly to the surface without subsequent release (static adsorption) of the enzyme. In dynamic adsorption, if the rate of LDH degradation by ZFO NPs is higher, then the LDH levels will decrease over time in a rapid manner, while in the static adsorption, the LDH molecules are unavailable for measurement due to the surface attachment [33]. Hence, after obtaining the results, it can be implied that due to the rise of ZFO NP concentrations there was a higher adsorption of LDH molecules onto their surface, thus decreasing the dose-response curve measurements by either dynamic or static adsorption. Furthermore, the toxicity of ZFO NPs previously shown in the earlier cytotoxicity assays suggests that at higher NP concentrations there was an increase in cell death, thus a higher LDH release occurred by the limited number of available cells. Therefore, the high NP concentration range during the exposure time could offer a bigger surface for dynamic or static adsorption for LDH molecules, thus reducing the viability of the cells. This can be interpreted from the release of LDH by the cells and then the LDH is adsorbed on the surface of the NPs, which could show false positive results. On the other hand, in the HEK-293 cell line, the dose-response curve exhibits an increase in the LDH release as the concentrations of ZFO NPs increases (Figure 7C). The plausible explanation as to why, in the 400 and 500 µg/mL, the amount of HEK-293 cell deaths is higher than in MDA-MB-231 cell line, which could be due to the NP absorption in the MDA-MB-231 cell line. So, these results suggest that LDH results must be read with caution, and they confirm the importance of linking several different cytotoxic assays for the safety and cytotoxic evaluation of NPs. Finally, these results confirm ZFO NPs are more toxic to the MDA-MB-231 and MCF-7 cancer cell line than HEK-293 normal cell lines.

Based on the results of all the three assays, it was corroborated that the there is no inimical effect of ZFO NPs on HEK-293 non-cancerous cell lines but there were many discrepancies with respect to both MDA-MB-231 and MCF-7 cancer cell lines sensitivity. It was quite difficult to determine the degree of cytotoxicity of ZFO NPs on each cancer cell line, i.e. Triple-negative breast cancer (MDA-MB-231) and breast adenocarcinoma cells (MCF-7). Since there were irregularities in dose-dependent cytotoxicity of cancer cell lines, there was a need of automated cell counting and sorting system such as flow cytometric analysis to determine which cancer cell line was more sensitive to ZFO NPs. 

### 2.5. Apoptosis Assay Using Flow Cytometry

In order to corroborate the good safety profile of benign cells (HEK-293) towards ZFO NP exposure, and to distinguish as to which breast cancer cell line (MDA-MB-231 and MCF-7) is more sensitive, flow cytometry studies were performed. Different concentrations of ZFO NPs (0, 8, 30, 50, and 250 µg/mL) were used for the flow cytometry-based assays. The two modes of cell death, apoptosis or necrosis, were measured using Annexin V-FLUOS (FITC) and propidium iodide (PI) dyes. The cellular uptake can be detected soon after 5–6 h, however, toxicity is generally seen at 24 h [38]. A two-dimensional contour density plot of early apoptosis (FITC) against late apoptosis (PI) was made. Each plot is separated into four quadrants and changes are measured based on no uptake or if there is uptake of either one or both the dyes. Every single cell will occupy one of the four spaces of the plot depending upon the process the cell is going through.

Cells not stained with either of the dyes, FITC and PI, are living cells. These cells are in the live cell quadrant of the contour plot (down left or Q3). Cells stained only with FITC dye are early apoptotic cells and are in the apoptotic cell quadrant of the plot (downright or Q4). Cells stained with both the dyes, FITC and PI, are late apoptotic cells (upper right or Q2) and necrotic cells (upper left or Q1). A total number of 20,000 cells were measured per ZFO NP concentration. Each cell was located in each quadrant depending on the status of the cell; live cells, early apoptosis, late apoptosis and necrosis (Table 1, Figure 8).

Moreover, the key point of this research was to identify if the non-cancerous cell line was susceptible to the treatment of ZFO NPs and as to which breast cancer cell line was more sensitive. All the different ZFO NP concentrations induced a higher percentage of the cell line, MDA-MB-231, in the quadrants of early apoptosis, late apoptosis and necrosis as compared to MCF-7 and HEK-293 (Figure 9). Late apoptosis and Q1 quadrant for the MDA-MB-231 cell line confirmed that this cell line is more sensitive towards the exposition of NPs with a higher percentage of dead cells as compared to the MCF-7 cell line after 24 h exposure (Figure 9).

Additionally, the non-cancerous cell line HEK-293 presented a lower percentage of cells in the quadrants of early apoptosis, necrosis, and late apoptosis in comparison to MDA-MB-231 (Figure 9). Hence, the flow cytometry results confirmed and correlated well with the scores from the MTT, CV and LDH assay, confirming triple-negative breast cancer cell line (MDA-MB-231) as more sensitive to several ZFO NP concentrations exposure. Also, HEK-293 exhibits a lower sensitivity to enter an early and late apoptotic process exhibiting a good-safety profile against non-cancerous cells. Finally, flow cytometry results also showed the resistance of the breast cancer cell lines (MCF-7) wherein very lower percentage of cells enter in late apoptotic phase as compared to MDA-MB-231 (Figure 9).

An early signature marker of apoptosis is the damage in the asymmetry of the phospholipid layer from the cell membrane resulting in the translocation of phosphatidylserine from inner face to the outer surface of the cell. The employed FITC binds specifically to the exterior ligands of the phosphatidylserine in the cell membrane forecasting an early apoptotic stage of the cell [39,40]. This grade of apoptosis from nanoparticle-exposed cells was quantified using the FITC dye in flow cytometry. Also, PI intercalates with the double strand of the nucleic acid during the excitation of blue light producing red fluorescence. However, PI cannot pass into the living cell when the cell membrane is intact. When the cells enter an apoptotic stage, the cell membrane becomes debilitated, and PI can enter the cell and stain the nucleic acids [39]. Therefore, both dyes can be used to discriminate between living and dead cells. Furthermore, several methods are employed for NP cytotoxicity and to determine the cell viability of cells exposed to different NP concentrations. The most common is the MTT assay which measures the metabolic activity of the cell. This method along with crystal violet and LDH assay generates a colored product, which can be easily quantified with spectrophotometric techniques [40]. Nevertheless, accurate quantification of NPs is not always precisely done since their optical properties can create an absorbance in an equivalent manner thus leading to a false estimation of the cell viability [40]. In addition, nanoparticles can have some interference in the colorimetric and absorbance cytotoxic assays, so the flow cytometry assay which is an automated cell counting and sorting system can give us an in-depth information and confirm the results of all the three cytotoxic assays such as MTT, crystal violet, and LDH. Additionally, with these experiments, the cell viability can be precisely quantified using flow cytometry-based assays since it is the cumulative output from thousands of individually measured cells. [40]. Therefore, flow cytometry has several advantages such as a low possibility of interference from NPs during the measurements, two different cell death mechanisms that can be differentiated as early and late apoptosis, and an understanding of the biochemical and molecular mechanisms of cell death after the exposure to different concentrations of magnetic nanoparticles [40]. This is a breakthrough since triple-negative breast cancer expresses a poor prognosis, is a more aggressive disease, has a low overall survival, has a mesenchymal phenotype, is multidrug resistance disease, is insensitive to treatments with antiestrogens, is more immunogenic, and has fewer options for treatments due to the lack of receptors or enough protein for hormone therapy. All of these restrict the therapies with radiation, immunotherapy, chemotherapy, and lastly with surgery [41,42]. Consequently, future treatments with ZFO NPs against triple-negative breast cancer could be an option since the results showed a good safety profile, a low benign cell death, and more sensitivity for this type of breast cancer.

### 2.6. Optical Microscopy Studies for Cell Morphological Analysis

Furthermore, the rigorous use of NPs in diagnosis and therapies in the medical field has led us to exploit such NPs to enter inside the cells based on their size. Once they enter, the NPs start accumulating within the cells, thus leading to cellular dysfunctions such as cytoskeletal damages as well as altered cellular metabolism by ROS mediation [43]. These cellular alterations are involved in protein and gene regulation, controlling cell migration, differentiation, proliferation, apoptosis, and tissue development [43]. Additionally, once the cytoskeleton is damaged, the focal adhesion of proteins and their following adhesion are affected, which begins beneath the filipodia (finger-like protrusions) and lamellipodia (branched actin filaments) [43]. In this study, the cytotoxic results of the different assays were confirmed by observing the morphology of the cells using digital photographs obtained by Nikon ECLIPSE TS100 microscope attached to the ProgRes^®^ CT3 at 20×. In the ProgRes^®^ CapturePro 2.10.0.1—JENOPTIK Optical Systems GmbH software, the images exhibit a reduction in the cell viability of HEK-293, MDA-MB-231, and MCF-7 as per the Figure 10A–C, respectively. Furthermore, it is clearly seen that the untreated cells were highly confluent, and they are spreading efficiently with the aid of lamellipodia and filipodia (as indicated by white circles in the figure). At 300 μg/mL, all cell lines exhibited stunted growth of lamellipodia and filipodia, and further, as the concentrations of ZFO NPs start increasing, especially at 600 μg/mL, all cell lines were demonstrated to possess much smaller cell area inhibiting the formation of lamellipodia and filipodia (as indicated by white circles in the figure). At that higher concentration, the cell morphology was completely damaged in comparison to the untreated ones and were completely round, causing cell shrinkage. There was higher dead cell debris observed in cancer cell lines in comparison to normal cell lines (as indicated by white circles in the figure). These results confirm that the lamellipodia and filipodia are vastly affected by the NPs, thus blocking their formation, leading to the decrement in the tumor dissemination and cancer invasion [43]. Ketebo and colleagues (2020) had the same result exhibiting an inhibition of filipodia and lamellipodia at 0.1 and 1 µg/L of silica-coated magnetic NPs with rhodamine B isothiocyanate [43]. Hence ZFOs provide the opportunity to generate a new and functional platform for Cancer Therapeutics.

## 3. Materials and Methods

### 3.1. Synthesis of ZFO NPs

5.2 gm of (0.5 M) of iron (III) chloride hexahydrate (FeCl_3_. 6H_2_O; Sigma-Aldrich 97%, Toluca, Mexico) and 2 gm (0.25 M) of zinc (II) chloride, granular (ZnCl_2_; J. T. Baker Analyzed ACS reagent, Radnor, PA, USA) were mixed in 25 mL of deoxygenated water under a nitrogen atmosphere. A total of 0.85 mL of 12.1 M hydrochloric acid (HCl; Sigma Aldrich) was added to the solution for 30 min under nitrogen atmosphere and stirred for 2 h at 85–90 °C. In addition, 15 gm of aqueous solution of 1.5 M sodium hydroxide pellets (NaOH; Meyer >97.0%) were added to 250 mL of deoxygenated water and mixed. Solutions of Fe (III), Zn (II), and HCl were added dropwise into the NaOH solution and mixed under magnetic stirring at 70 °C for 1 h under N_2_ atmosphere. The synthesis is based on the precipitation and the obtained precipitates were recollected and centrifuged for 20 min at 4000 rpm. The supernatant was thrown, and the precipitate was washed with 50 mL of deionized water. This procedure of washing was repeated three times. At the end of the last wash, the precipitated ZFO NPs were dried at 80–100 °C for 4 h. [2].

### 3.2. Characterization

The as-synthesized ZFO NPs were characterized using a BK-UV 1800 pc (BIOBASE, Jinan, Shandong, China) UV-vis spectrometer between 200 and 800 nm. Raman spectrum from 100 to 2000 cm^−1^ was acquired with a Xplora (Horiba, Palaiseau, France) spectrometer with an exciting radiation of 638 nm, power of 20–25 mW, objective ×10, and a CCD camera used as a detector. FTIR spectra were recorded with an IRA-affinity-1S spectrophotometer (SHIMADZU, Kyoto, Japan) and the LabSolution IR software. 5 μL of the NP ZFO NPs samples were added on a module ATR Specac Quest with a diamond prism and measured between the wavenumber interval of 400 cm^−1^ and 4000 cm^−1^, with a resolution of 4 cm^−1^. X-ray diffraction (XRD) analysis of the crystalline phase of ZFO were performed with a X-ray diffractometer D8 Advance (Bruker, Karlsruhe, Germany) with a Cu Kα lamp (Measurements parameters: radiation wavelength, 1.54 A; acceleration voltage, 40 KV; current 40 mA; using a coupled method). The obtained results were compared with the Joint Committee Powder Diffraction Standards (JCPDS) database for phase affinity. Transmission electron microscopy (TEM) analyses were performed with a FEI, Titan G2 80–300 instrument operated at 300 kV as acceleration voltage, equipped with an EDAX energy-dispersive X-ray spectroscopy (EDXS) detector. The ZFO NPs sample was also characterized by X-ray photoelectron spectroscopy in a Thermo-Scientific, K-Alpha spectrometer using AlKα radiation (E = 1.5 keV). With this technique we could record high resolution Fe2p, Zn2p, and O1s spectra to revise the chemical state of Fe, Zn, and O in the synthesized sample. The magnetic properties of the powdered NP ZFO sample were measured in a Quantum Design, MPMS3 magnetometer from the recording of isothermal field-dependent magnetic moment curves [m(H)] at 5 and 312 K, using a magnetic field interval between −70 and 70 kOe. In addition, zero field-cooled (ZFC) and field-cooled (FC) temperature-dependent magnetic moment curves [m(T)] were obtained for a temperature interval between 4 and 312 K at 100 Oe.

### 3.3. Cell Culture and MTT, Crystal Violet, and LDH Cytotoxic Assay

Human epithelial breast cancer cell line (MDA-MB-231) and breast cancer cell line (MCF-7) were obtained from Instituto Nacional de Ciencias Médicas y Nutrición Salvador Zubirán (INCMNSZ), Mexico, and human embryonic kidney 293 cell line (HEK-293) was obtained from Universidad Nacional Autónoma de México (UNAM). Both cell lines were cultured with high glucose Dulbecco’s modified Eagle’s medium (DMEM) (Sigma Aldrich, St. Louis, MO, USA) with 10% fetal bovine serum (Sigma Aldrich, USA) and incubated at 37 °C, 5% CO_2_. The 17,500 MDA-MB-231, 20,000 MCF-7 and 6000 HEK-293 cells were plated in wells of 48 well-tissue culture plates. The cells were kept for recovery for one day after the addition of the cells, and on the third day, they were stimulated. ZFO NPs were added in different concentrations in triplicates, as well as doxorubicin hydrochloride (Doxylid 50, CelonLabs, Nagpur, India) was added as a positive control, and untreated cells served as negative controls. ZFO concentrations were taken as follows: 4, 12, 20, 50, 100, 200, 300, 400, 500, and 600 μg/mL in MTT and crystal violet and 4, 50, 100, 200, 300, 400, 500, and 600 μg/mL in LDH assay. The treated cells were incubated at 37 °C, 5% CO_2_, for 48 h.

MTT assay: After the incubation period with the ZFO NPs and doxorubicin, the media was aspirated, then the wells washed with 1× phosphate-buffered saline (PBS) solution, and 300 μL of MTT (Sigma-Aldrich 98%) was added to every well, mixed gently and later incubated for 4 h at 37 °C, 5% CO_2_. The MTT assay in each well was carefully removed and replaced with 700 μL ethanol and mixed gently to dissolve the formazan crystals. 100 µL of the resolubilized formazan of each well was placed in a 96-well tissue culture plate and read in a plate reader at 570 nm (Thermo Scientific MULTISKAN Sky, Marsiling Industrial Estate Road 3, Singapore). The OD at 570 nm for the blue-purple formazan was used to evaluate the anti-proliferative effect of compounds. The OD at 570 nm for control cells were taken as 100% viability.

Crystal violet assay was performed directly in the 48-well tissue culture plate by adding 250 µL of 5% crystal violet stain solution and then incubated in an orbital shaker at 100 rpm for 5 min, at room temperature. The stain was removed, and the wells were washed four times with tap water and then the plates were left to dry. A total of 500 µL of a 10% *v*/*v* acetic acid solution was then added to each well and incubated in an orbital shaker at 100 rpm for 5 min, at room temperature. A total of 200 µL was removed and transferred to a 96-well tissue culture plate to read the optical density in a plate reader at 590 nm.

The Lactate dehydrogenase (LDH) cytotoxicity assay was performed as is described in In Vitro Toxicology Assay Kit, Lactic Dehydrogenase based, reference TOX7-1KT (Sigma-Aldrich), which can be explained as follows: ss mentioned above in the MTT and Crystal violet assay, after the incubation of the nanoparticles and Doxorubicin with the cell cultures and the medium, aliquots of the medium were removed. Later Lactate Dehydrogenase (LDH) assay mixture was added to the medium that was removed for the analysis. The medium volume was twice the LDH mixture. The plates were covered by an aluminum foil to avoid any interference of the light and the incubation is performed for 20–30 min. After the termination of the reaction using 1N HCl, the absorbance of the solution can be measured spectrophotometrically at 490 nm.

Flow cytometry-based assay was performed with a BD LSRFortessa^TM^ (BD Biosciences, San José, CA) flow cytometer containing a blue laser 488 nm laser, forward scatter (FSC) diode detector, and a side scatter (SSC) detector was used. MDA-MB-231, MCF7, and HEK-293 cell lines were seeded in 12 plate-wells for 24 h and treated with ZFO at different concentrations (0, 8, 30, 50, and 250 µg/mL) for 24 h at 37 °C. Cells without nanoparticles were considered as the control group. All plates were removed from DMEM and washed with 1 mL of PBS. The 1 mL of versene was added into each well of the plate and incubated for 10 min, then removed from each well and added into an independent Falcon tube with 1.5 mL of previously added PBS. Each Falcon tube was centrifuged for 3 min at 1500 rpm, resuspended with 1 mL of binding buffer, and centrifuged for 5 min at 4000 rpm. Lastly, each tube was resuspended with binding buffer with propidium iodide (Sigma-Aldrich) and Annexin V-FLUOS (Roche; 11828681001) and incubated for 30 min. All data analyses from 20,000 cells per sample were carried out in FACSDiva v. 8.1 flow cytometer (BD Bioscience) unless otherwise stated. Samples were gated as exhibited in representative example Appendix A.

## 4. Conclusions

In the present study, ZFO NPs are synthesized using a co-precipitation method. The results of the different techniques used for the characterization show that ZFO NPs were single phase with normal spinel structure. The crystallite size was determined by XRD and HR-TEM analysis, which was approximately 5 nm and showed patterns of successful synthesis with peaks related to the cubic phase. The UV-vis absorbance peaks exhibited three characteristic electronic transitions peaks of ZFO, thus confirming the presence of ZFO. Additionally, FTIR and RAMAN spectra presented bands related to Zn-O and Fe-O bonds. XPS analysis confirmed the presence of Zn, Fe, and O. Moreover, the observed maximum magnetization of ZFO powders was at 45.5 emu/g at 70 kOe. Finally, during the cytotoxic assays, MTT, crystal violet, LDH, and flow cytometry, some interactions could be noticed but all the assays showed zinc ferrite NPs damages more the cancerous cell lines MDA-MB-231 and MCF-7 than human embryonic kidney cell line HEK-293. These results were confirmed with the digital photographs showing the lamellipodia and filipodia inhibition as the ZFO NPs concentration raised. These studies will help in optimizing cytotoxicity assays for future nanoparticle-based drug-delivery, contrast agent, biolabeling, and Theranostics.

## Figures and Tables

**Figure 1 ijms-24-12860-f001:**
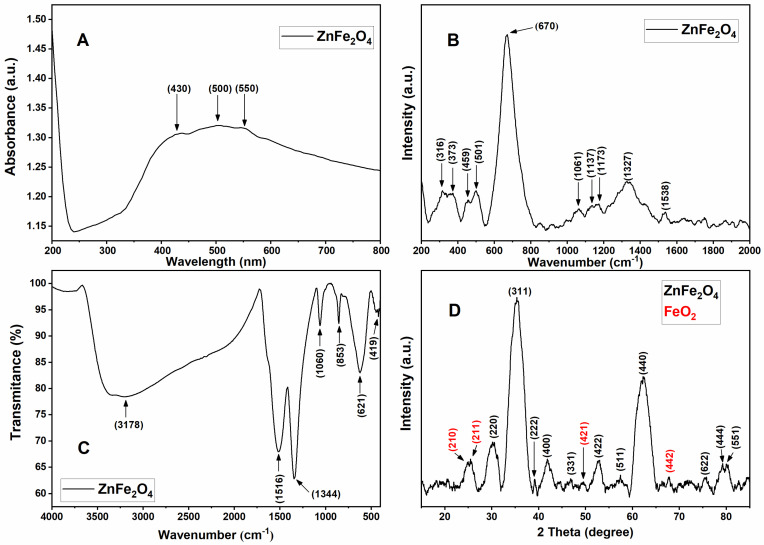
(**A**) UV-visible spectra of zinc ferrite nanoparticles exhibiting three absorption peaks at 430, 500, and 550 nm that correspond to three different transitions. (**B**) Raman spectra of ZFO NPs at room temperature from 200 to 2000 cm^−1^. (**C**) FTIR spectrum of zinc ferrite nanoparticles. (**D**) X-ray diffractogram of zinc ferrite nanoparticles exhibiting crystalline planes of (220), (311), (222), (400), (331), (422), (511), (440), (622), (444), and (551) from the cubic system of zinc ferrite JCPDS pattern no. 01-089-4926 and 00-022-1012 as well as crystalline planes of (210), (211), (421), and (442), according to JCPDS pattern no. 00-039-1346 that corresponds to iron oxide (Fe_2_O_3_) + Fe^3+^ → O^−^ + Fe^2+^ indirect transition [10]. Likewise, the obtained dark brown color of ZFO proved the visible light absorption capacity of these NPs [8,10].

**Figure 2 ijms-24-12860-f002:**
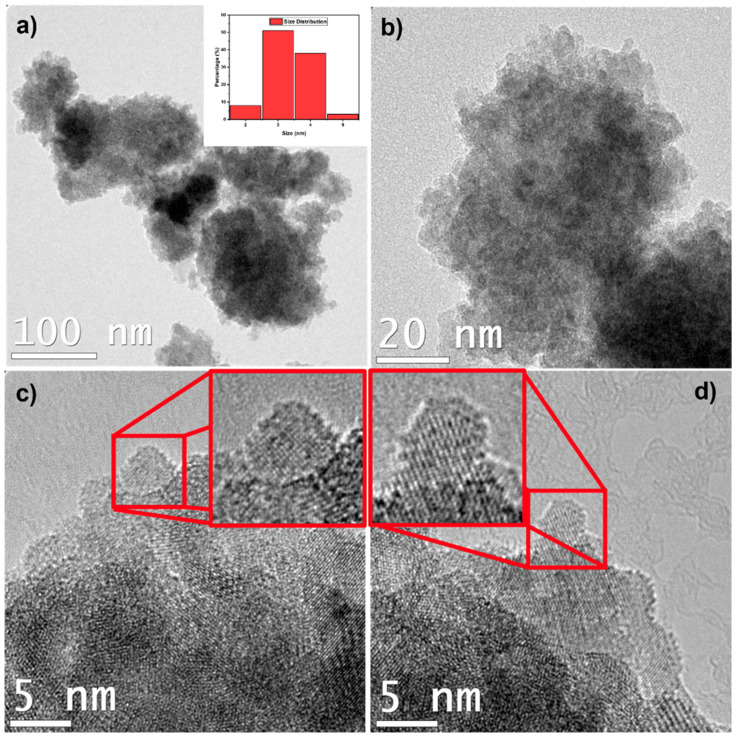
Transmission electron microscopic images of the ZFO NPs nanoparticles: (**a**,**b**) images showing the agglomerates formed by the nanoparticles. Figure (**a**) shows an inset of the histogram of the size distribution of the nanoparticles. (**c**,**d**) High-resolution transmission electron microscopic images (HR-TEM) of the ZFO NPs. Insets: HR-TEM images of individual nanoparticles.

**Figure 3 ijms-24-12860-f003:**
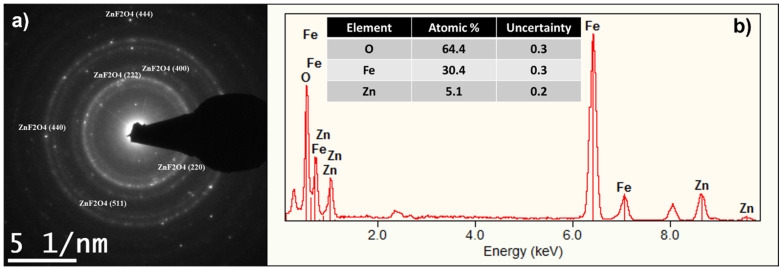
(**a**) SAED pattern from the ZFO NPs nanoparticles. (**b**) EDXS spectra showing the composition of the nanoparticles.

**Figure 4 ijms-24-12860-f004:**
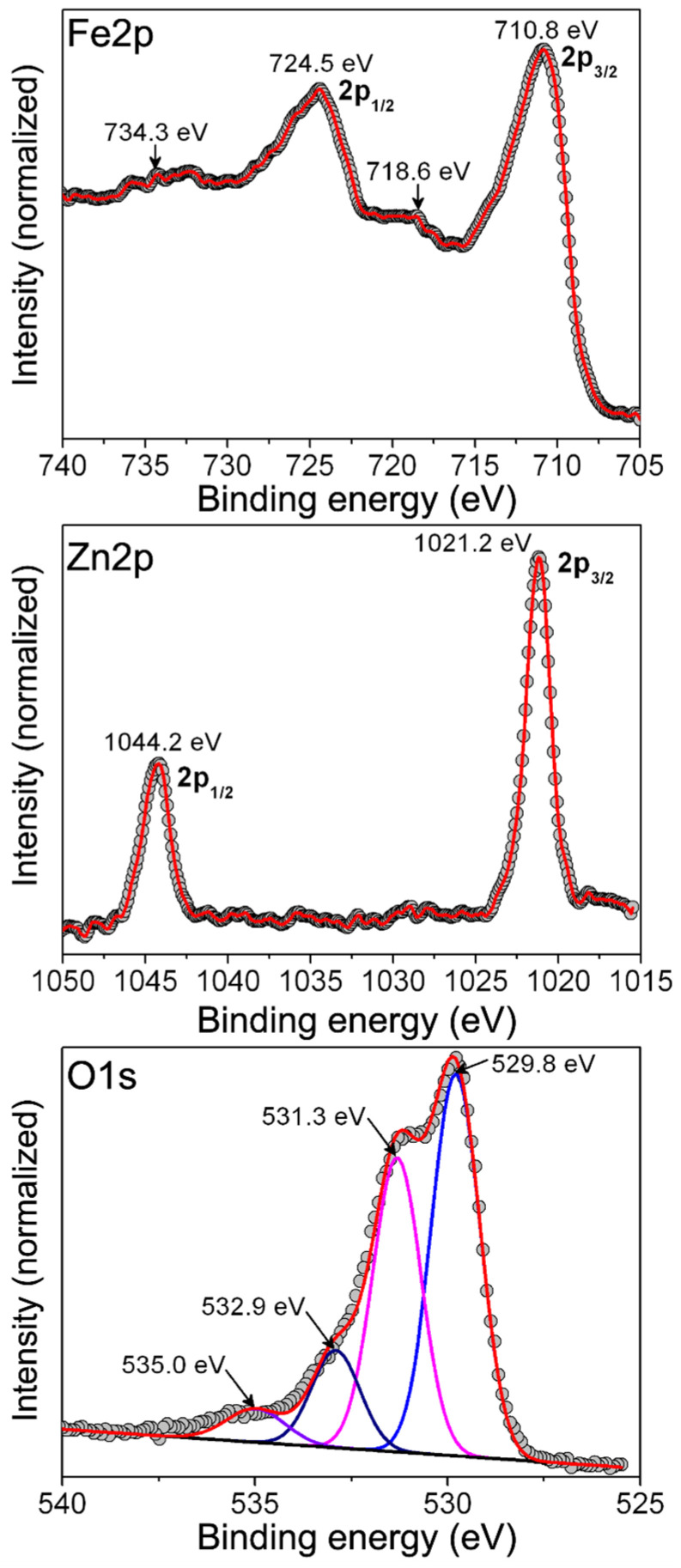
X-ray photoelectron spectroscopy showing high resolution Fe2p, Zn2p, and O1s spectra recorded for the powdered ZFO sample.

**Figure 5 ijms-24-12860-f005:**
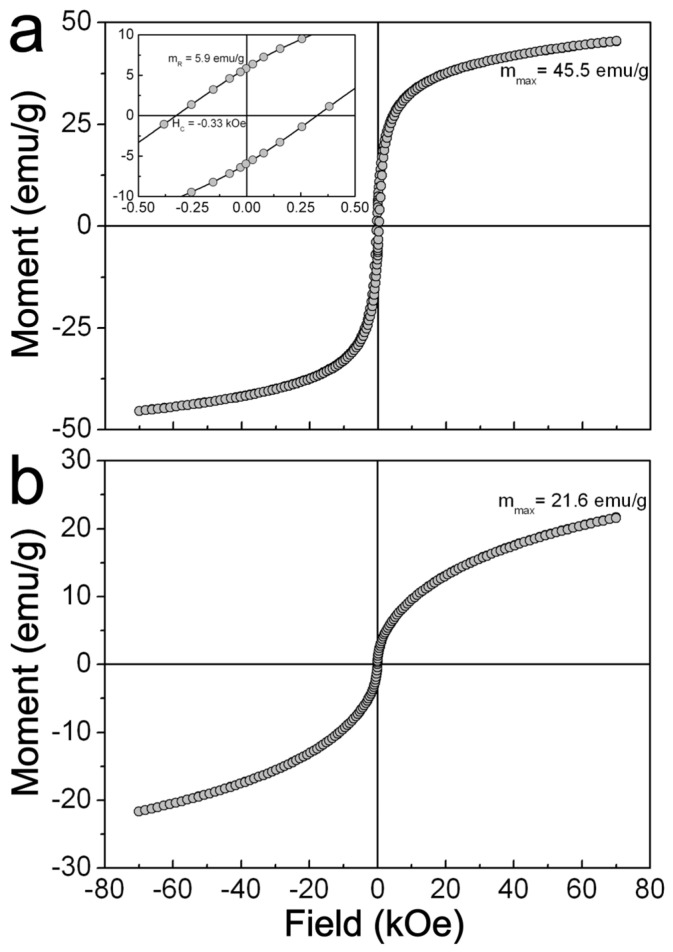
Isothermal field-dependent magnetic moment curves recorded for ZFO NPs sample at: (**a**) 5 K; (**b**) 312 K. The axes of the plot in the inset have the same units as seen in the plots (**a**,**b**).

**Figure 6 ijms-24-12860-f006:**
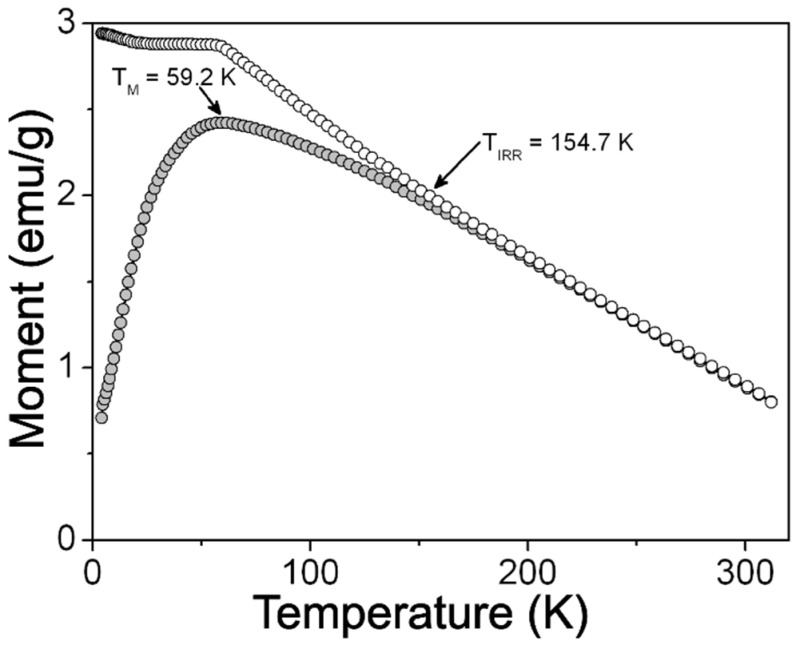
Zero field-cooled (ZFC) (gray circles) and field-cooled (FC) (white circles) temperature-dependent magnetic moment curves measured for ZFO NPs sample at 100 Oe.

**Figure 7 ijms-24-12860-f007:**
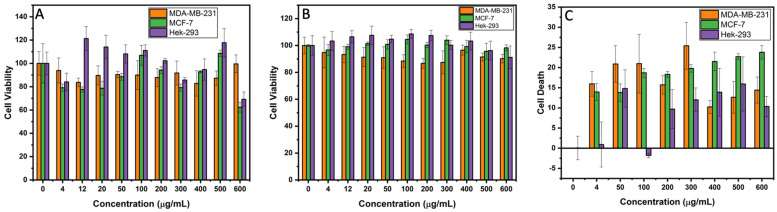
(**A**) Cytotoxicity profile of ZFO NPs using a MTT assay of MDA-MB-231, MCF-7, and HEK-293 cell lines. (**B**) Cytotoxic profile of Crystal violet assay of MDA-MB-231, MCF-7, and HEK-293 cell lines using ZFO NPs. (**C**) Dose-response data of LDH release from MDA-MB-231, MCF-7, and HEK-293 cells after the ZFO NP exposure.

**Figure 8 ijms-24-12860-f008:**
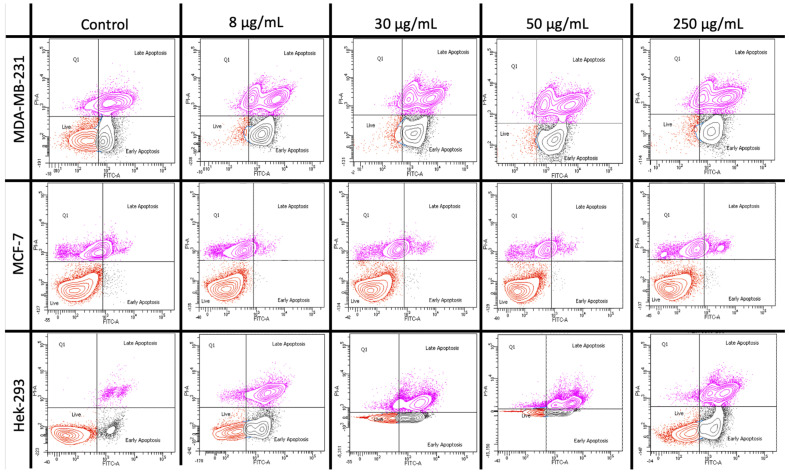
Representative two-dimensional contour density plots to determine fraction of live, early apoptosis, and late apoptosis of cells (MDA-MB-231, MCF-7, and HEK-293) when exposed to different concentrations (0, 8, 30, 50, and 250 µg/mL) of zinc ferrite nanoparticles for 24 h. Early and late apoptosis were measured by using Annexin V-FLUO and propidium iodide dyes.

**Figure 9 ijms-24-12860-f009:**
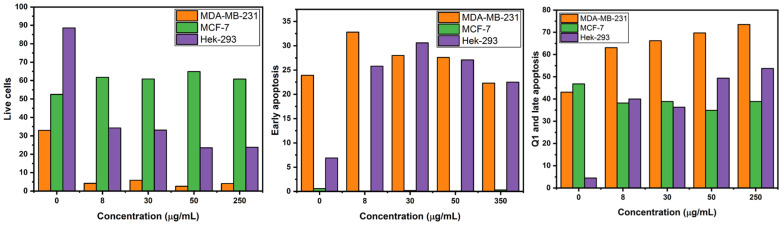
Percentages of live cells and cells entering in early and late apoptosis of MDA-MB-231, MCF-7, and HEK-293 in a flow cytometry study.

**Figure 10 ijms-24-12860-f010:**
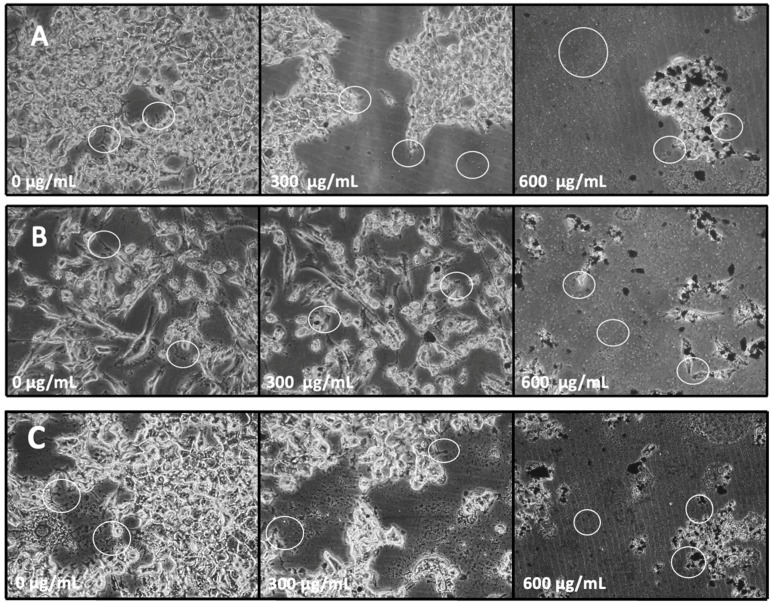
(**A**) Digital microscopic images of HEK-293 cell line, (**B**) Digital microscopic images of MDA-MB-231 cell line, and (**C**) Digital microscopic images of MCF-7 cell line after the exposure of 0 μg/mL, 300 μg/mL, and 600 μg/mL of ZFO NPs.

**Table 1 ijms-24-12860-t001:** Obtained percentages from integration of cells (MDA-MB-231, MCF-7, and HEK-293) in each quadrant (live, early apoptosis, and late apoptosis) normalized to the total cell number 20,000 when exposed to different concentrations of zinc ferrite nanoparticles (0, 8, 30, 50, and 250 µg/mL).

	Live (%)			Early Apoptosis (%)			Q1 and Late Apoptosis (%)		
Concentration	MDA-MB-231	MCF-7	HEK-293	MDA-MB-231	MCF-7	HEK-293	MDA-MB-231	MCF-7	HEK-293
0 µg/mL	33	53	89	24	1	7	43	47	5
8 µg/mL	4	62	34	33	0	26	63	38	40
30 µg/mL	6	61	33	28	0	31	66	39	36
50 µg/mL	3	65	24	28	0	27	70	35	49
250 µg/mL	4	61	24	22	0	23	74	39	54

## Data Availability

Data shared are in accordance with consent provided by participants.

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
