# Peer review of "A Comparative and Critical Analysis for In Vitro Cytotoxic Evaluation of Magneto-Crystalline Zinc Ferrite Nanoparticles Using MTT, Crystal Violet, LDH, and Apoptosis Assay"

_ijms, 2023, doi:10.3390/ijms241612860_

Round 1
Reviewer 1 Report
After reading the manuscript, I am afraid that the level of novelty presented within this study is not sufficient to grant acceptance and publication in IJMS. In brief, authors have repeated already published method of ZFO NPs synthesis, and performed very vague cytotoxicity screening (with exceedingly high NP concentrations).
Moderate English editing is required.
Author Response
We are very thankful for your kind suggestions. This has helped us to enhance the article further. Please see the attachment

Reviewer 2 Report
The manuscript by Juan Luis de la Fuente-Jiménez et al. nicely characterizes ZNPs. While the characterization of the ZNPs is extensive, interesting and mostly complete, the in vitro application of these ZNPs is partially weak. Therefore, the authors should consider the following comments:
Overall, the Figure display could be significantly improved. Why not generating multiple-panel figures? This would increase the readability of the MS significantly and would also improve data representation.
The title of the MS is misleading, as the authors include the term TNBC. However, only one cancer cell line and one non-cancer cell line is analyzed in the MS. Therefore, to strengthen the MS, further (breast) cancer cells should be investigated in response to ZNP treatment. Possibly, the terms “triple-negative breast” will subsequently be removed from the title, as the effect is TNBC unspecific and also other cancer cells are affected.
A suggestion to complete the characterization of the ZnFe NPs would be, to incorporate the different spectra/analyses for isolated Zn and Fe ions, to give a better understanding, whether the spectra are a sum of the ion composition of the Zinc ferrite nanoparticles or whether the spectra are specific for its chemical structure.
The authors should include statistics/ histograms for the size distribution of the NPs in Fig.5.
The authors claim, that ZNPs exhibit their cytotoxic effects by stimulating mitochondrial ROS production. Therefore, ROS should be assessed to verify/ confirm this assumption/ statement.
In the abstract, the authors claim that the ZNPs exhibit the cytotoxic effect on TNBCs. This is true, as the authors observe a more pronounced effect on the viability of MDA-MB-231 cells compared to HEK293 cells. However, to strengthen the statement, other TNBC cells should be tested. Is the effect specific for TNBCs? What about other (breast) cancer cells?
The authors should include statistics in Figures 10-12. Are the differences significant?
Despite there might be a more pronounced effect on the viability of cancer cells compared to non-cancer cells, the effect/difference is small, with highest effects at 200-300 µg/mL and death rates of ~20-25%. What about co-treatments with established anti-cancer drugs? Are the effects potentiated/ synergistic? Furthermore, the authors should include respective information in the discussion, that a further optimization of the ZNP composition/ complexation with other drugs/ cotreatments may improve cytotoxicity/ specificity.
The authors performed crystal violet assays by reading CV absorbance. Mostly, however, CV staining is analyzed by colony counting/ assessment. By absorbance readings, ZNPs may interfere with absorbance/ CV staining of the cells. Therefore, colony formation assays and images of respective plates should be performed and added in the MS to validate the findings. These experiments would furthermore demonstrate a potential impact of ZNPs on colony development/ growth/ survival.
The authors claim that the effect of the ZNPs is specific for the (TNB) cancer cells. This statement is weak, as only one cancer and one non-cancer cell line was investigated. Where do the differences come from? Do these cells show differential ZNP uptake? Further cell lines should be included, and the authors should analyze the intracellular ZNP concentration after cell treatment, which may indicate a different uptake efficacy of the cells. As the authors have access to electron microscopy, such experiments might be performed using EM to see intracellular ZNP distribution/ analyze the concentration/cell.
What should the circles in Figure 13 indicate?
The quality of english is fine and only requires minor editing, as only some spelling mistakes and additional spaces were present.
Author Response
We are thankful for the kind suggestions by the reviewer and please see the attachment for point-by-point response.

Reviewer 3 Report
The MS ´Optical, morphological, magnetic, and structural characteriza- 2 tion of Zinc ferrite nanoparticles for comparative cytotoxicity 3 evaluation with different viability assays on Triple-negative 4 Breast cancer cells´.
It is very interesting and I like to read it.
The results of the synthesis of the nanomaterial are well presented and characterized by various tools.
I find difficulty in understanding the cytotoxicity part. it would be easier to understand if some more explanation is added.
Figure 13 also not easy to visualize the changes as claimed in the text. So I like to suggest adding schematics figures along with the microscopic figures to understand the changes and visualize them. If possible please add a high-resolution or magnified version of the figures.
it would also be good if add some more applications and future prospects in the conclusion part.
Author Response

(The authors gave the same response as above.)

Reviewer 4 Report
Dear Authors and Editor,
The subject presented in the manuscript is very important and it can be published after logical explanation of a certain obscurity.
Authors should explain non-regularity of influence of concentrations ZFO NPs on MDA-MB-231 and 383 Hek-293 cell lines (Figures 10 – 11). Under increasing of concentration cell viability is increasing (about 200 milg/mL). This non-regularity put in doubt the presented biological studies and, of cause, conclusions.
Quality of Raman spectra should be improved.
Resolution of the IR and Raman measurements should be presented.
English should be improved.
Author Response

(The authors gave the same response as above.)

Reviewer 5 Report
This is a well-designed and organized research article to emphasizing the Zinc ferrite nanoparticles (ZFO NPs) are a promising platform for nanomedicine-based cancer theranostics. The results are rich and support the hypothesis. My minor comments are as follows:
1. In introduction mention that MDA-MB-231 is a triple-negative breast cancer cells and cite the reference.
2. Please mark * to indicate whether statistical significance exists between the two groups from Figure 10-12. I find that the authors mentioned “These results confirm that the ZFO NPs damage more cancerous MDA-MB-231 cell lines than the normal Hek-293 cell lines” in figure 11.
3. Figure 12 is not clear. Is the vertical axis the number of cell deaths or the LDH concentration? It would be better to the show the correlation of LDH concentration and cell death.
4. The limitations of current study
Author Response

(The authors gave the same response as above.)

Round 2
Reviewer 1 Report
Overall, manuscript has been improved. I recommend that authors split section 2.4 into three subsections related to each viability assay. In that way, the readability of the manuscript will be improved. However, manuscript still lacks novelty to be published in IJMS.
Minor edits are needed.
Reviewer 2 Report
The manuscript by Juan Luis de la Fuente-Jiménez et al. nicely characterizes ZNPs. The MS improved in terms of representation. However, the authors ormitted all suggested experiments and rather raised concerns, especially about the validity of MTT, Crystal Violet and LDH assays, than confirming their findings in other (breast cancer) cell lines.
Why not giving Numeric distribution (histrogram) of NP size?
Many different assessment methods for ROS are available. If one is logistically unavailable, why not using another one?
The authors state that there might be interference with the ZNPs and Crystal Violet absorbance reading. Hence, the performance of "Standard colony formation assays" is even more required and important to validate their findings.
As the authors only demonstrate one cell line as a responder, no conclusions can be made on the effect of the ZNPs on ((breast) cancer) cells. The effect might be random on MDA-MB-231 cells and not be reproduced in other (TNBC) cells....
Based on the poor characterization of the effect of the ZNPs on cells and the mentioned other issues, I need to suggest a rejection of the article at the current stage. Once the authors added additional experiments, a reconsideration might be appropriate.
Round 3
Reviewer 2 Report
The authors adressed most comments properly.
One last aspect that may require the attention of the authors is the title of the manuscript, as the term "triple negative breast cancer cells" is incorrect. On the one hand, MCF-7 do not respond to the treatments, however, several manuscripts also consider MCF-7 cells as triple negative. Furthermore, also HEK293 cells were tested (which are rather somatic than cancer cells), which showed a response to the treatments. Hence, I would recommend to change the title to "Optical, morphological, magnetic, and structural characterization of Zinc ferrite nanoparticles for comparative cytotoxicity evaluation with different viability assays on cell lines"
Minor grammatical / stylistic check may be required.
Author Response
1. One last aspect that may require the attention of the authors is the title of the manuscript, as the term "triple negative breast cancer cells" is incorrect. On the one hand, MCF-7 do not respond to the treatments, however, several manuscripts also consider MCF-7 cells as triple-negative. Furthermore, also HEK293 cells were tested (which are rather somatic than cancer cells), which showed a response to the treatments. Hence, I would recommend to change the title to "Optical, morphological, magnetic, and structural characterization of Zinc ferrite nanoparticles for comparative cytotoxicity evaluation with different viability assays on cell lines"
Answer: Based on the suggestions of the editor as well as the kind reviewer, we have changed the title of the manuscript which is:
A comparative and critical analysis for in vitro cytotoxic evaluation of magneto-crystalline Zinc ferrite nanoparticles using MTT, Crystal violet, LDH, and apoptosis assay
2. Minor grammatical / stylistic check may be required.
We have done corrections to eliminate the minor grammatic errors in the manuscript